# A Validated IVRT Method to Assess Topical Creams Containing Metronidazole Using a Novel Approach

**DOI:** 10.3390/pharmaceutics12020119

**Published:** 2020-02-03

**Authors:** Seeprarani Rath, Isadore Kanfer

**Affiliations:** 1Division of Pharmaceutics, Faculty of Pharmacy, Rhodes University, Grahamstown 6139, South Africa; seeprarath81@gmail.com; 2Leslie Dan Faculty, University of Toronto, Toronto, ON M5S 3M2, Canada

**Keywords:** IVRT, metronidazole, topical cream, semisolid dosage forms, sameness, FDA’s SUPAC-SS guidance, acceptance criteria, positive and negative controls, discriminatory ability

## Abstract

An IVRT method was developed and validated to confirm its reproducibility, precision, sensitivity, selectivity, accuracy, robustness, and reliability. A novel approach was used to demonstrate the appropriateness of the IVRT method to accurately assess “sameness” between topical products and to confirm that the methodology applied also possesses the requisite discriminatory power to detect differences should such differences exist between products. In the first instance, the reference product (Metrocreme^®^) containing 0.75% metronidazole (MTZ) was tested against itself as a positive control, to accurately demonstrate “sameness”, where the results met the relevant acceptance criteria falling within the limits of 75–133.33% in accordance with the FDA’s SUPAC-SS guidance. In addition, two specially prepared creams containing 25% less and 26% more MTZ, i.e., 0.563% and 0.945%, served as negative controls and were compared against the reference product. Neither of these creams fell within the “sameness” acceptance criteria, thereby confirming the discriminatory ability of the IVRT method to detect differences between MTZ products. Furthermore, another cream containing 0.75% MTZ tested against the reference product was shown to be pharmaceutically equivalent to the reference product. These results confirm the appropriateness of the IVRT method as a valuable tool for use in the development of topical MTZ products intended for local action and indicate the potential for general use with other topical products.

## 1. Introduction

In vitro release testing (IVRT) entails measurement of the drug released from the vehicle into a receptor medium, separated by an inert membrane [1] and used to quantify the amount of active pharmaceutical ingredient (API) released from semisolid dosage forms and to determine its release rate [2]. The application of IVRT to assess “sameness” of semisolid dosage forms following minor changes related to an approved topical dosage form was described in the United States Food and Drug Administration’s (FDA’s) SUPAC-SS guidance [3]. Furthermore, IVRT is a USP compendial method used for performance testing of semisolid dosage forms [4] and is also an extremely useful tool in product development [5]. It has been proven to be useful to detect differences in qualitative (Q1) and quantitative (Q2) properties between similar products and also in the microstructure and arrangement of matter between formulations (Q3) [6]. In 2003, the experts from FIP/AAPS pointed out that there was an absence of a standard in vitro test that could be applied to semisolid dosage forms [7]. Until recently, however, amongst the numerous publications involving the application of IVRT, comprehensive validation of IVRT systems and methods have been conspicuously absent from the literature. Although various efforts [8,9,10] have been made by several researchers to develop a standardized method to measure in vitro drug release from a product using diffusion cells, a comprehensive validation that would be generally applicable to all topical dermatological dosage forms has only recently been published [11]. This is essential to ensure that the developed IVRT method and the associated system has the necessary discriminatory capabilities to determine “sameness” between products and also, more importantly, differences, which may affect clinical performance.

Since clinical endpoint studies have been made mandatory by most of the regulatory authorities to establish the safety and efficacy of generic topical products except in the case of topical corticosteroid products wherein a vasoconstrictor assay (VCA) or human skin blanching assay (HSBA) is accepted in accordance with the US FDA’s guidance, an appropriate IVRT method could provide valuable and compelling information to justify waivers of bioequivalence studies (biowaivers) for topical semisolid products intended for local action. In this respect, it is interesting to note that the US FDA [12,13,14,15] and the European Medicines Agency (EMA) [16] have published guidances that recommend the use of IVRT for such purposes. It is interesting to note that the abovementioned FDA guidances are product-specific and the acceptance criteria are based on those recommended in the USP [4], in accordance with the FDA’s SUPAC-SS guidance [3], whereas the acceptance criteria in the recent EMA draft guideline [16] are more stringent. Furthermore, the EMA draft guideline [16] describes the validation conditions and requisite discriminatory power of the method to support claims of therapeutic equivalence with a reference product instead of undertaking clinical trials.

In view of the fact that market approval of a generic topical metronidazole (MTZ) product requires a clinical endpoint study in patients to show safety and efficacy of such products, an IVRT method was developed to investigate “sameness” and differences between a test formulation and the market-approved reference product, Metrocreme^®^. The primary objective, therefore, was to investigate the feasibility of using a nonclinical approach, such as IVRT, for consideration as a biowaiver to obtain market approval. Consequently, an attempt was made to develop and comprehensively validate an IVRT method which has the necessary discriminatory power to detect “sameness” and differences in topical cream products containing 0.75% MTZ. A novel approach of incorporating a positive control wherein the reference product was compared to itself and two negative controls, i.e., products containing <25% or >25% potency compared to the reference product, was implemented.

## 2. Materials and Methods

### 2.1. Materials

#### 2.1.1. Chemicals

MTZ was obtained from Sigma-Aldrich and stored in a cool, dry area that was free from light exposure. HPLC-grade methanol (200 UV ROMIL—SpSTM Super Purity Solvent) was obtained from Romil Ltd. (Waterbeach, Cambridge, UK). The water used for chromatography was prepared by reverse osmosis, followed by filtration through a Milli-Q system (Millipore, Bedford, MA, USA).

#### 2.1.2. Formulations

Metrocreme^®^ (Galderma Laboratorium GmbH, Dusseldorf, Germany) containing 0.75% MTZ was used as the reference MTZ cream. Creams containing 0.375%, 0.75% (T_1_), and 1.125% MTZ were specially manufactured to validate the IVRT method. In addition, creams T_2_ and T_3_, which contained 25% less (0.563%) and 26% more (0.945%) MTZ than the reference product were used as negative controls at the lower and upper acceptance limits, respectively, in accordance with the SUPAC-SS limits [3], to establish the utility of the developed and validated IVRT method to assess inequivalence. Test cream (T_1_) was also used to assess “sameness” for the purpose of marketing a generic MTZ cream.

#### 2.1.3. Equipment

Mettler^®^ Model AE 135 analytical balance (Mettler^®^ Inc, Zurich, Switzerland) and MX5 Mettler^®^ Toledo Microbalance (Mettler^®^ Inc, Zurich, Switzerland) were used for weighing samples and standards, respectively. Micropipettes P100 and P1000 (Pipetman™, Gilson^®^, Villiers-le-Bel, France) were used to transfer standard and sample solutions for dilutions.

The concentration of hydrocortisone in PVT samples was analyzed on a Waters Alliance HPLC system that was equipped with a separation module (Model 2695) and a PDA detector (Model 2996) and Empower^®^ 3 data acquisition system (Waters, CT, USA). The chromatographic separation was achieved by using a Phenomenex Luna^®^ C18 (2) 5 μm (150 × 4.6 mm) column (Torrance, CA, USA) maintained at 25 °C. The mobile phase consisted of acetonitrile: water (30:70 *v*/*v*). A flow rate of 1 mL/min was maintained, and a detection wavelength of 254 nm was used. Samples were injected at ambient temperature during analysis.

The concentrations of MTZ were determined by using a Waters Acquity UPLC system equipped with a photo diode array detector (PDA), Empower^®^ 3 data acquisition system (Waters, Milford, MA, USA). The chromatographic separation was achieved by using an Acquity UPLC BEH C18 1.7 µm (2.1 × 100 mm). A mobile phase of methanol/water (40/60 *v*/*v*) was pumped at a flow rate of 0.2 mL/min, and the eluate was monitored at a wavelength of 318 nm. Samples (2 μL) were injected at ambient temperature during analysis. 

In vitro release studies were performed by using a Hanson VDC system (Hanson Research Corporation, Chatsworth, CA, USA) consisting of six vertical diffusion cells (Volume: 7.9 mL, orifice: 15 mm) with closed cell tops, mounted on a six-station LGA platform and a magnetic stirrer (Variomag^®^, Berkley, CA, USA), and a Control Unit (Telemodul 40S H+P Labortechnik GmbH, Munich, Germany), connected to a PolyScience circulating water bath (Niles, IL, USA). The diffusion cells and apparatus were assembled with donor and receptor chambers that were separated by a chosen membrane. The receptor chamber was filled with 7.9 mL of SABAX Pour Saline (0.9% NaCl solution, Adcock Ingram Critical Care (Pty) Ltd., Midrand, South Africa), which served as the receptor fluid and maintained at 32 °C (to mimic the physiological temperature). The receptor medium was continuously stirred, using individual magnetic stirrer bars, coupled with helical mixer springs, in each of the VDCs. One-milliliter syringes (Terumo^®^ (Philippines) Corporation, Laguna, Philippines) with sampling needles (21G x 3.5” Terumo^®^ Spinal Needle, Terumo^®^ Corporation, Tokyo, Japan) were purchased from local pharmacies.

### 2.2. Methods

#### 2.2.1. UPLC Method Validation

The UPLC method was validated according to ICH guidelines and relevant criteria, as described by Tiffner et al. [11].

#### 2.2.2. Apparatus Qualification 

Factors that can influence the drug release were assessed to comply with the requirements of an apparatus qualification test in accordance with the USP procedures for apparatus qualification for the VDC system [4,17]. Environmental factors such as suitable working area, workbench levelness, no direct exposure to sunlight and/or direct cooling vents, and failure-resistant power supply to all the electronic components were ensured. The system was placed on a dedicated sturdy wooden workbench with a free distance of more than 76 cm above the system. Additionally, the orifice diameters and capacities of each of the VDCs, the temperature of the receptor medium, stirrer speed, and the dispensed sample volume were validated. 

#### 2.2.3. Performance Verification Test (PVT)

The PVT was performed by using 1% hydrocortisone cream (Emo-Cort^®^, GlaxoSmithKline Inc., Mississauga, ON, Canada) applied on Tuffryn membranes [17,18]. Two IVRT runs were performed, each with six VDCs in parallel, by stirring the receptor medium at 600 rpm and maintaining the receptor medium temperature at 32 ± 1 °C. Prior to the commencement of the experiment, the receptor chambers were filled with degassed receptor medium consisting of ethanol: water (30:70 *v*/*v*) and VDC system were allowed to equilibrate at 32 ± 1 °C for approximately 30 min. The cream was accurately weighed (~300 mg) and applied evenly onto the relevant membranes, which had been presoaked in the receptor medium for 30 min. Aliquots of 200 µL were withdrawn from the receptor chambers of each of the 6 VDCs at 1, 2, 3, 4, 5, and 6 h, and the VDCs were subsequently replenished with 200 µL of receptor medium after each withdrawal.

#### 2.2.4. MTZ Solubility and Receptor-Fluid Selection

Various receptor fluids such as SABAX Pour Saline (0.9% NaCl solution, Adcock Ingram Critical Care (Pty) Ltd., Midrand, South Africa), lactate buffer pH 4.5, phosphate buffer pH 4.5, phosphate buffer pH 6.8, water/ethanol (50:50 *v*/*v*), phosphate buffer pH 4.5: ethanol (50:50 *v*/*v*), and 0.9% NaCl solution/ethanol (50:50 *v*/*v*) were investigated. The solubility of MTZ in these receptor fluids was evaluated in triplicate by placing 200 mg MTZ in 10 mL of receptor fluid to yield a saturated solution. The solution was stirred for 6 h at 600 rpm and 32 ± 1 °C and allowed to stand at 32 ± 1 °C overnight [11]. Aliquots of the supernatant were withdrawn, filtered, diluted, and then analyzed on a UPLC, to determine the concentration of dissolved MTZ. The receptor medium should be able to dissolve >10-fold maximum expected concentration of the drug. 

#### 2.2.5. Membrane Screening 

Membrane screening was carried out by using various synthetic membranes: Magna Nylon (0.45 µm, 25 mm, GVS Life Sciences, Sanford, ME, USA); Tuffryn (0.45 μm, 25 mm, Pall Corporation, Ann Arbor, MN, USA); HVLP (0.45 μm, 47 mm, Merck Millipore Ltd., Ireland); Cellulose acetate (0.45 μm, 25 mm, Sartorius, Göttingen, Germany); Mixed Cellulose Ester (0.45 μm, 47 mm, Advantec MFS, Inc., Pleasanton, CA, USA), and Strat-M^®^ (25 mm, Millipore, MA, USA). Binding of MTZ to the membranes was investigated by immersing individual membranes (*n* = 3) in 10 mL of 0.9% sodium chloride solution containing MTZ at 32 ± 1 °C, for 6 h. The test solution without any immersed membrane, prepared in triplicate, was allowed to equilibrate for 6 h at 32 ± 1 °C and was used as a control. The MTZ concentrations in all the solutions were determined by using the UPLC method described above. Recoveries were calculated for each of the test solutions containing the membranes by comparing them with the control solution. The mean percent recovery for each membrane should be within ±5% [11].

#### 2.2.6. Sampling Duration

IVRT runs were performed by varying the duration of the runs and the sampling intervals. Three runs were performed—6 h IVRT run with sampling at every 30 min; a 3 h run with sampling at every 15 min; and a 1.5 h run with sampling at every 15 min. The sampling duration with desired linearity (*R*^2^ > 0.95) and recovery (≤30%) was considered based on Higuchi’s assumptions [19,20]. At least 6 sampling intervals were used.

#### 2.2.7. IVRT Method

Each of the IVRT runs was conducted by using six VDCs in parallel, and the receptor medium was stirred at 600 rpm, while maintaining the temperature at 32 ± 1 °C. Prior to the commencement of the experiment, the receptor chambers were filled with degassed 0.9% NaCl solution, and the VDC system was allowed to equilibrate at 32 ± 1 °C, for approximately 30 min. The cream was accurately weighed (~300 mg) and applied evenly onto the Nylon membranes, which had been presoaked in the receptor medium for 30 min. Aliquots of 200 µL were withdrawn from the receptor chambers of each of the 6 VDCs every 15 min for 90 min, and the VDCs were subsequently replenished with 200 µL of receptor medium after each withdrawal. The stirring was stopped during sample withdrawal and immediately resumed once the aliquots were withdrawn and receptor media replenished in all the VDCs. Care was taken that the duration between the stopping and resumption of the stirring was less than 1.5 min. The aliquots were analyzed by using the UPLC method described above.

#### 2.2.8. Calculation of Release Rates

The periodical concentrations of MTZ measured using a UPLC method were used to determine the amount of API released. The Higuchi model, which assumes the existence of perfect sink conditions, was used to determine the release rates, using Equation (1). Dilution of the receptor medium due to replacement of the sampled amount was taken into account, and the concentrations of MTZ in the receptor medium (*C_n_*) at different sampling times were calculated, using Equation (1).
(1)$$Q_{n}=C_{n}\frac{V_{c}}{A_{c}}+\frac{V_{s}}{A_{c}}\overset{n}{\underset{i=1}{\sum }}C_{i-1}$$
where*Q_n_* = amount released at time (*n)* per unit area in µg/cm^2^;*C_n_* = concentration of drug in receptor medium at different sampling times (*n*) in µg/cm^3^;*V_s_* = volume of the sample in cm^3^;*V_c_* = volume of the cell in cm^3^;*A_c_* = area of the orifice of the cell in cm^2^.

In accordance with Higuchi’s square root approximations [20] given by Equation (2), a plot of *Q* vs. $\sqrt{t}$ will be linear with a slope of $\sqrt{2ADC_{s}}$.
(2)$$Q=\sqrt{2ADC_{s}t}$$
where*Q* = the amount absorbed at time *t* per unit area of exposure µg/cm^2^; *A* = the concentration of drug expressed in µg/cm^3^; *C_s_* = the solubility of the drug in µg/cm^3^;*D* = the diffusion constant of the drug molecule. 

Hence, the release rate corresponds to the slope of the regression line of the plot of *Q_n_* vs. $\sqrt{t}$. *Q_n_* is affected by sample volume, VDC volume, and by the diameter of the orifice of the VDC. Hence, these parameters were carefully evaluated during apparatus qualification [11,17].

#### 2.2.9. Validation of the IVRT Method

The IVRT system was validated in accordance with the method described by Tiffner et al. [11]. Metrocreme^®^ (Galderma Laboratorium GmbH, Düsseldorf, Germany) containing 0.75% MTZ, being an RLD, was used as the reference MTZ cream. Cream products containing 0.375%, 0.75%, and 1.125% MTZ were used to establish the sensitivity, selectivity, and specificity of the IVRT method. 

Differences in release rates between the creams containing different concentrations of MTZ were evaluated to determine the sensitivity of the method.

Specificity of the method was assessed by investigating proportionality of the MTZ release rate to the MTZ concentration in the test creams. A box and whisker plot was prepared to estimate the coefficient of determination (*R*^2^). The specificity of the method can be confirmed if a specifically proportional linear relationship can be established between the formulations of different strengths, i.e., *R*^2^ > 0.9.

The statistical approach for product “sameness” testing described in the USP general chapter <1724> [4] was applied to analyze all results. The selectivity of the IVRT method was investigated by determining the release rates of the lower and higher strengths of MTZ test creams (i.e., 0.375% and 1.125%, respectively) and compared with the release rates of the 0.75% MTZ test cream. Pairwise comparisons of MTZ release rates in three IVRT runs performed as a part of linearity, precision, and reproducibility investigations were evaluated in order to establish whether the reference, Metrocreme^®^, 0.75% MTZ was equivalent to itself. 

The effect of two temperature variations (−2 and +2 °C) relative to the nominal temperature of 32 °C (at 32 ± 1 °C) and also two stirring-rate variations (−60 rpm and +60 rpm) relative to the nominal stirring rate of 600 rpm were investigated in order to determine the robustness of the method to minor perturbations, using the reference product, Metrocreme^®^, 0.75% MTZ. 

Dose depletion was assessed by determining the amounts of MTZ released from the cream during the IVRT runs and recovered in the receptor solution. The recovery was calculated by dividing the average cumulative amount released at the last time point (*t* = 90 min) by the amount of MTZ present in the applied dose (~2.25 mg) by using data from each of the 18 VDCs within the 3 IVRT runs. 

#### 2.2.10. Comparative IVRT of MTZ Cream Products

In order to confirm the utility of the developed and validated IVRT method for assessment of “sameness” and differences between MTZ creams, comparative IVRT runs were conducted with the test (T_1_, T_2_, and T_3_) and the reference cream. The testing was carried out in accordance with the FDA’s SUPAC-SS guidance [3]. Initially, two IVRT runs were carried out to compare T_1_ and the reference, such that the arrangement enabled the measurement of drug release from each of the creams in all six cells at the end of the experiment. Following this, three IVRT runs were carried out to compare creams T_2_ and T_3_ with the reference, to serve as negative controls in such a way that the drug release from each of the creams was measured in all six cells at the end of the experiment. Additionally, the two IVRT runs conducted as a part of IVRT validation served to determine whether the method can accurately detect “sameness”, wherein the reference product was compared to itself to serve as a positive control.

The cumulative amount of MTZ released per unit area was plotted against the square root of time and the release rates were determined. A total of six release rates per cream were obtained. Pairwise comparison was carried out by computing the ratios of release rates (T/R ratios) and establishing the 90% confidence interval (CI), using the Mann–Whitney U Test, where the 90% CI should lie within the limits of 75–133.33% to confirm “sameness” between products, in accordance with the FDA’s SUPAC-SS guidance [3]. 

## 3. Results

### 3.1. Validation of the UPLC Method and Qualification of the IVRT System 

The developed UPLC method was validated for the analysis of MTZ in IVRT samples, and the results are summarized in Table 1, according to the acceptance criteria defined in the ICH guidelines. The results for qualification of the IVRT system are described in Table 2. 

### 3.2. Performance Verification Test

The results obtained from the PVT runs are summarized in Table 3. 

### 3.3. Receptor-Fluid Selection

The observed solubilities of MTZ were >2848.10 µg/mL in all the receptor fluids, which is >10 times the maximum expected concentration.

### 3.4. Membrane Screening

Average percentage recoveries of 98.47 ± 0.81, 98.07 ± 2.09, 97.84 ± 1.27, 97.70 ± 1.63, 96.80 ± 1.97, and 91.04 ± 3.28 for Nylon, Tuffryn, Cellulose acetate, Strat-M, HVLP, and Mixed Cellulose Ester, respectively, were obtained. 

### 3.5. Sampling Duration

The release rates and coefficients of determination (*R*^2^) for different times are shown in Table 4. Significant differences between release rates by using different sampling durations were observed by using the F-test, which generated *p*-values > 0.05.

### 3.6. Validation of the IVRT Method

The results obtained from the three IVRT runs of 0.75% MTZ cream confirmed a linear relationship (*R*^2^ ≥ 0.99) between the amount of drug released and the square root of time. 

The mean release rates for 0.375%, 0.75%, and 1.125% MTZ creams were 19.17 μg/cm^2^/min^1/2^, 40.04 μg/cm^2^/min^1/2^, and 66.28 μg/cm^2^/min^1/2^, respectively, and a linear relationship (*R*^2^ = 0.9994) was observed between the release rates of the different strengths of MTZ cream (Figure 1).

The data to assess selectivity are depicted in Table 5, where the 90% CI for the reference product, Metrocreme^®^, 0.75% MTZ, against itself, fell within the acceptance criteria of 75–133.33% [3], whereas that for the 0.75% MTZ cream against the 0.375% and 1.125% MTZ creams fell completely outside the 75–133.33% limits.

The mean release rates obtained for the robustness runs were 45.67 μg/cm^2^/min^1/2^ for 30 °C, 600 rpm; 47.83 μg/cm^2^/min^1/2^ for 34 °C, 600 rpm; 47.02 μg/cm^2^/min^1/2^ for 32 °C, 540 rpm; and 48.19 μg/cm^2^/min^1/2^ for 32 °C, 660 rpm.

The mean recoveries for IVRT runs conducted as a part of linearity, precision, and reproducibility were found to be 30.27 ± 0.52%, 29.00 ± 0.60%, and 29.98 ± 0.81% respectively, confirming that the dose depletion was within the specified range of ≤30%. 

### 3.7. Comparative IVRT of MTZ Topical Cream Products

A comparison between the reference and test product T_1_ is depicted in Figure 2. The mean release rates were 37.80 ± 2.01 µg/cm^2^/min^1/2^ and 32.89 ± 1.60 µg/cm^2^/min^1/2^ for the reference and T_1_, respectively. The calculated upper and lower limits for the pairwise comparison were 81.72% and 91.24%, which fell within the CI limits of 75–133.33% (Appendix A).

The comparison between the reference and the test products T_2_ and T_3_ is depicted in Figure 3. The mean release rates were 38.47 ± 1.35 µg/cm^2^/min^1/2^ for the reference and 27.42 ± 0.99 µg/cm^2^/min^1/2^ and 51.20 ± 0.60 µg/cm^2^/min^1/2^ for the test creams T_2_ and T_3_, respectively. 

The calculated upper and lower limits for the pairwise comparisons were outside the CI limits of 75–133.33% for both test creams (Table 6). The CI limits for the positive control were well within the acceptance criteria (75–133.33%), as shown in Table 6.

## 4. Discussion

The developed UPLC method, validated for the analysis of MTZ in IVRT samples in accordance with the ICH guidelines, met the acceptance criteria for all the validation parameters described by Tiffner et al. [11]. It was found to be selective, linear over a range of 0.5–100 μg/mL, accurate, precise, robust, sensitive with an LLOQ of 0.5 μg/mL, and offering sample stability for seven days. All the parameters essential for qualification of the IVRT system met the predefined acceptance criteria as described by Tiffner et al. [11], with an exception of the VDC volume. The mean capacity of the VDCs did not conform to the specifications provided by the manufacturer. However, it was precise and suitable for conducting IVRT experiments. Hence, the subsequent IVRT experiments were conducted by using these cells, and the calculations were tailored accordingly.

The results obtained from the PVT runs duly complied with the predefined acceptance criteria, and the low variabilities observed during the runs confirm the suitability and reproducibility of the IVRT system for its relevant application.

Although the observed solubilities of MTZ were >10 times the maximum expected concentration, as recommended [9], 0.9% NaCl solution was chosen as the receptor medium for subsequent studies, as it more appropriately reflects physiological conditions and is inexpensive and readily available. All the membranes except the Mixed Cellulose Ester membrane showed an acceptable recovery, indicating low MTZ binding. The membranes were therefore confirmed to be inert and would not act as a rate-limiting barrier for MTZ. Because of its higher-percentage recovery and availability, Nylon was chosen as the membrane of choice for this study.

The release profiles obtained from the IVRT runs with different sampling durations and intervals showed significant differences in MTZ release rates. It was observed that the release rates decreased considerably with time, whereas the recovery increased, resulting in curvature in the release profile. The profiles showed a deviation from linearity after 90 min. This is in agreement and closely correlated with the Higuchi’s square root approximations [19,20]. Deviation from linearity of release of API from different semisolid dosage forms may be due to differences in physicochemical properties of such formulations. This can be explained by considering the square root approximations developed by Higuchi, which presumes that the percent of API released from the applied dose is ≤30%. The Higuchi equation describes the initial 30% release from topical formulations under perfect sink conditions as linear. Since the ultimate objective of this study was not to evaluate complete release from the cream but to assess the “sameness” between two creams containing 0.75% MTZ, an IVRT method spanning over a duration of 90 min, with a sampling interval of 15 min, was chosen in accordance with the Higuchi’s approximation (recovery ≤30%).

The inter-run and intra-run variabilities were <15%, confirming that the IVRT method was precise and reproducible. The mean release rate for 0.75% MTZ cream was higher than that of 0.375% MTZ cream and lower than that of 1.125% MTZ cream, indicating that the method was sensitive to the changes in MTZ concentrations. The linear relationship (Figure 1) indicated that the release rate proportionately increased with the increase in MTZ concentration, demonstrating the specificity of the developed IVRT method. The IVRT method has the necessary ability to determine “sameness” as the 90% CI for the reference product, Metrocreme^®^, 0.75% MTZ, against itself, fell within the acceptance criteria of 75–133.33% [3]. Furthermore, the ability of the method to assess differences was confirmed as the 90% CI for the 0.75% MTZ cream against the 0.375% and 1.125% MTZ creams fell completely outside the 75–133.33% limits. From the results obtained, it is evident that the method also has the discriminatory ability to detect differences in MTZ concentrations between the respective products. This outcome was expected, as the creams (0.375%, 0.75%, and 1.125%) differed only in terms of Q2 properties (i.e., the amount of MTZ) and contained the same ingredients and also were manufactured using the same process. Q1 and Q3 properties were therefore constant for all the creams. The developed IVRT method is thus sensitive to the changes in the formulations specifically related to differences in Q2. Minor differences in stirring rates (±60 rpm) or temperature (±2 °C) did not significantly affect MTZ release rates, i.e., they did not deviate from the nominal values by more than 15%. These results suggest that the developed IVRT method possesses the necessary properties to confirm robustness.

Since it is important to confirm the discriminatory ability of the IVRT method, both positive and negative controls were included to indicate “sameness” and differences, if any, in terms of the FDA’s SUPAC-SS acceptance criteria [3]. Interestingly, acceptance criteria of 90–111% with a 90% CI was recommended in the recent EMA draft guideline [16]. This increased stringency is somewhat questionable. The reference product was used as a positive control when compared to itself during validation studies, whereas, for different strengths of the MTZ creams at concentrations below (T_2_) and above (T_3_) the acceptance limits were used as negative controls. Accordingly, a comparison between the reference and test product T_1_ indicated “sameness” between the two creams, thereby confirming that the IVRT method has the requisite properties.

Based on the results from the positive control studies, the ability of the IVRT method to indicate “sameness” was confirmed. Furthermore, data using negative controls indicated the capability of the IVRT method to detect inequivalence between MTZ creams.

## 5. Conclusions

The apparatus qualification and PVT of the IVRT system duly complied with the necessary requirements to assess the release of API from topical cream products. The low variabilities observed during the PVT runs confirm the suitability and reproducibility of the IVRT system for its relevant application. An IVRT method was carefully developed and comprehensively validated to assess the release of MTZ from cream products, taking into account the various parameters that may affect the API release rate. The validation procedures were carried out in accordance with the recommendations in both the FDA guidance [3,4] and EMA draft guideline [16]. It was then successfully applied to assess “sameness” for the purpose of marketing a generic MTZ cream, wherein a 0.75% MTZ cream was compared with the reference, Metrocreme^®^, 0.75% MTZ. Furthermore, the inclusion of a positive and negative controls was a novel approach to demonstrate the discriminatory power of the IVRT method. The positive and negative controls complied with the necessary acceptance criteria for equivalence and inequivalence, respectively, confirming the discriminatory power of the IVRT method. This is in accordance with the SUPAC-SS acceptance criteria [3] for a biowaiver, where generic products containing <25% or >25% potency compared to the reference product are deemed inequivalent. The resulting data indicate that the developed IVRT method can be applied to accurately and precisely assess “sameness” and differences (Q2) between MTZ cream products as a valuable procedure in formulation development. Furthermore, this method has potential for use as a tool to obtain a biowaiver for MTZ creams.

## Figures and Tables

**Figure 1 pharmaceutics-12-00119-f001:**
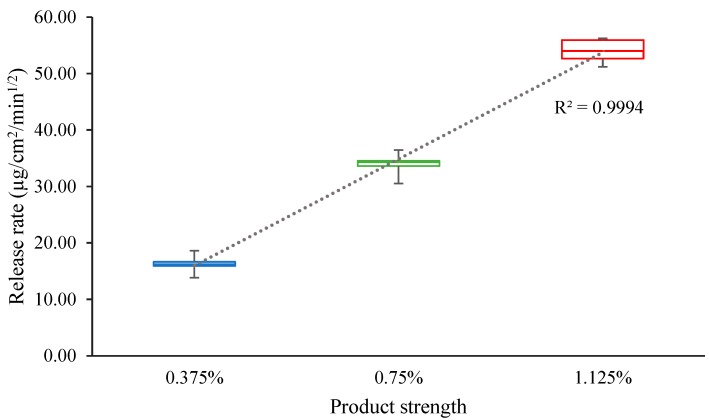
Box and whisker plot of the measured release rates for the three test MTZ creams, with concentrations of 0.5% (blue), 1% (green), and 1.5% (red).

**Figure 2 pharmaceutics-12-00119-f002:**
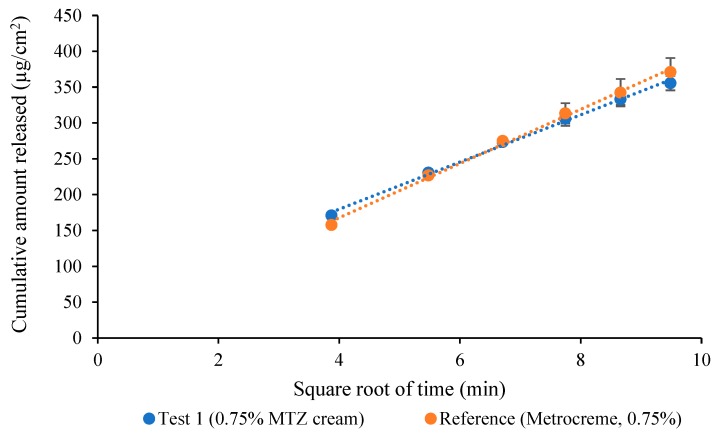
Release-rate curves from 0.75% MTZ cream (T_1_) vs. Metrocreme^®^, 0.75% MTZ (R).

**Figure 3 pharmaceutics-12-00119-f003:**
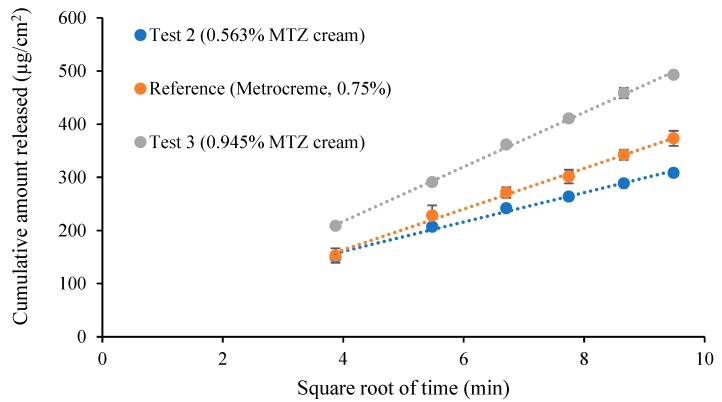
Release-rate curves from 0.563% and 0.945% MTZ cream (T_2_ and T_3_) vs. Metrocreme^®^, 0.75% MTZ (R) (*n* = 6).

**Table 1 pharmaceutics-12-00119-t001:** Predefined acceptance criteria and the results obtained from the UPLC validation for IVRT method [11].

Parameter	Acceptance Criteria	Results	Pass
Selectivity and specificity	${\bar{RT}}_{p}$ − ${\bar{RT}}_{pm}$ < 10% $IC_{n}$ = 0 µg/mL, $IC_{nm}$ = 0 µg/mL	0.063% $IC_{n}$ = 0 µg/mL, $IC_{nm}$ = 0 µg/mL	Yes
Linearity	75% of the standards meet the following criteria: For 5–100 μg/mL: $IC_{meas,lin}$ ∈ [$IC_{nom}$ ± 15%] For 0.5 μg/mL: $IC_{meas,lin}$ ∈ [$IC_{nom}$ ± 20%] *R*^2^ ≥ 0.950	30 out of 30 standards (100%) met the acceptance criteria *R*^2^ ≥ 0.999	Yes
Accuracy	For $IS_{100}$ and $IS_{50}$: $\bar{IS_{meas,ac}}$ ∈ [$IS_{nom}$ ± 15%] For $IS_{5}$: $\bar{IS_{meas,ac}}$ ∈ [$IS_{nom}$ ± 20%]	0.69% and 1.48% for $IS_{100}$ and $IS_{50}$ 2.08% for $IS_{5}$	Yes
Precision	Intra-day CV < 15% for $IS_{100}$ to $IS_{50}$ Intra-day CV < 20% for $IS_{5}$	1.30%, 0.63%, and 0.66% for $IS_{100}$ 1.03%, 0.93%, and 0.11% for $IS_{50}$ 1.07%, 6.29%, and 7.35% for$IS_{5}$	Yes
	Inter-day CV < 15% for $IS_{100}$ to $IS_{50}$ Inter-day CV < 20% for $IS_{5}$	6.00% for $IS_{100}$ 0.72% for $IS_{50}$ 1.07% for $IS_{5}$	Yes
Robustness	Inter-run CV < 15% for $IS_{100}$ to $IS_{50}$ Inter-run CV < 20% for $IS_{5}$	10.03% for $IS_{100}$ 9.05% for $IS_{50}$ 9.27% for $IS_{5}$	Yes
Stability	For $IS_{100}$ and $IS_{50}$: $\bar{IS_{meas,ac}}$ ∈ [$IS_{nom}$ ± 15%] For $IS_{5}$: $\bar{IS_{meas,ac}}$ ∈ [$IS_{nom}$ ± 20%]	Bench-top after 7 days: 0.22% for $IS_{100}$ 0.36% for $IS_{50}$ 5.01% for $IS_{5}$ UPLC machine after 7 days: 0.23% for $IS_{100}$ 0.29% for $IS_{50}$ 0.45% for $IS_{5}$ Refrigerator after 7 days: 0.34% for $IS_{100}$ 0.11% for $IS_{50}$ 0.49% for $IS_{5}$	Yes
LLOQ, LOD	-	LLOQ: 0.5 µg/mL LOD: 0.167 µg/mL	

*RT_p_* = retention time of MTZ in positive control (min). *RT_pm_* = retention time of MTZ in matrix positive control (min). *IC**_n_* = concentration in negative control (µg/mL). *IC_nm_* = concentration in matrix negative control (µg/mL). *IC_meas,lin_* = measured concentration during the linearity run (µg/mL). *IC_nom_* = nominal concentration (µg/mL). *IS_meas,ac_* = measured average concentration of the spiked solutions in the accuracy run (µg/mL). *IS_nom_* = nominal concentration of the spiked solutions (µg/mL). *IS**_x_* = concentration of the spiked samples where *x* is concentration in µg/mL.

**Table 2 pharmaceutics-12-00119-t002:** Predefined acceptance criteria and the results obtained from apparatus qualification for the VDC system.

Parameter	Acceptance Criteria		Results		Pass
	Mean ± Tolerance	Range of Variation	Mean	Range of Variation	
Capacity of the cells	7.00 ± 0.35 mL	≤0.21 mL	7.92 mL	0.02 mL	No
Diameter of the orifice	15.00 ± 0.75 mm	≤0.45 mm	14.98 mm	0.10 mm	Yes
Temperature of the receptor medium	32 ± 1 °C	-	32.07 °C	0.05 °C	Yes
Speed of the magnetic stirrer	600 ± 60 rpm	≤12 rpm	601.67 rpm	2.53 rpm	Yes
Dispensed sampling volume	200 ± 10 µL	-	203.67 µL	4.58 µL	Yes
Bench top levelness	Not more than 1°		<1°		Yes

**Table 3 pharmaceutics-12-00119-t003:** Predefined acceptance criteria and results of the performance verification test (PVT).

Parameter	Acceptance Criteria	Results
Intra-run variability	Intra-run CV for the first run (*n* = 6 VDCs) < 15%	8.13%
	Intra-run CV for the second run (*n* = 6 VDCs) < 15%	12.85%
Inter-run variability	Inter-run CV for both runs (*n* = 12 VDCs) < 15%	10.30%
Product “sameness” testing	The 90% CI should fall within the limits of 75–133.33%	Lower limit: 87.82% Upper limit: 116.52%

**Table 4 pharmaceutics-12-00119-t004:** Differences in parameters observed by using various sampling duration (mean ± SD).

Time (min)	Release Rate (µg/cm^2^/min^1/2^)	*R*^2^ Value	Recovery (%)
90	39.47 ± 2.30	0.998 ± 0.003	29.90 ± 1.20
180	34.19 ± 1.97	0.988 ± 0.006	38.32 ± 1.19
360	27.58 ± 2.09	0.994 ± 0.002	42.75 ± 3.57

**Table 5 pharmaceutics-12-00119-t005:** Computed 90% CI to assess the selectivity of the IVRT method.

Pairwise Comparison *	Computed 90% CI	
	Lower Limit	Upper Limit
0.375% MTZ cream vs. 0.75% MTZ cream	45.10	52.47
1.125% MTZ cream vs. 0.75% MTZ cream	152.04	167.79
Metrocreme^®^, 0.75% MTZ (run 1) vs. Metrocreme^®^, 0.75% MTZ (run 2)	93.61	100.22
Metrocreme^®^, 0.75% MTZ (run 1) vs. Metrocreme^®^, 0.75% MTZ (run 3)	87.40	97.66
Metrocreme^®^, 0.75% MTZ (run 2) vs. Metrocreme^®^, 0.75% MTZ (run 3)	93.06	99.76

* The release rates used for pairwise comparison are provided under Appendix A.

**Table 6 pharmaceutics-12-00119-t006:** Computed 90% CI for negative and positive controls used to assess discriminatory power of the IVRT method.

Pairwise Comparison *	Computed 90% CI		Decision
	Lower Limit	Upper Limit	
**Positive control** Metrocreme^®^, 0.75% MTZ vs. Metrocreme^®^, 0.75% MTZ	93.61	100.22	Pass
**Negative controls**			
0.563% MTZ cream vs. Metrocreme^®^, 0.75% MTZ	68.25	74.21	Fail
0.945% MTZ cream vs. Metrocreme^®^, 0.75% MTZ	128.36	137.11	Fail

* The release rates used for pairwise comparison are provided under Appendix A.

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
