# Peer review of "A Validated IVRT Method to Assess Topical Creams Containing Metronidazole Using a Novel Approach"

_pharmaceutics, 2020, doi:10.3390/pharmaceutics12020119_

Round 1
Reviewer 1 Report
In my opinion, this paper is mainly a case-study to validate an IVRT method according to the criteria prescribed by the own authors in a previous paper. I am not capable of judging its experimental aspects. On the other hand, as a non-specialist, I consider that it is written in a clear and understandable way. My review focuses on the statistical aspects of the paper and specially in the in vitro release equivalence tests.
The authors seem to apply correctly all the statistical directives in the 1997 SUPAC-SS. But, as they do not provide the raw data, I am not capable of securing this point. For that reason and also because this paper may constitute an interesting case-study, the authors should provide the original datasets and more detail on how they performed the statistical analyses.
A second point is that the 1997 SUPAC-SS guidance has more than 20 years. The 2018 EMA draft guidance on topical products (cited but not considered at all by the authors), and also some recent literature considering the FDA case, may imply some changes in the near future (e.g., other acceptance limits) on how these analyses must be performed. This paper is not a specific submission asking for FDA approval, quite the contrary, it may constitute a useful methodological contribution if some of these possibilities were considered in the statistical analyses and in the discussion.
Reviewer 2 Report
This is a well designed study that has carefully followed FDA guidance for carrying out IVRT studies and comes after a similar 2018 paper dealing with IVRT of acyclovir semisolid products. This latter paper already has been cited 7 times and so there appears to be interest in this area.
There is little to criticize in this paper. "Sensitivity" occurs twice in the list of features in the abstract (lines 9/10). For the benefit of readers who may not be familiar with it, I recommend showing the Higuchi relationship explicitly when you describe how the release rate is derived (lines 187-188). It will then be clear why release rate corresponds to the slope of the regression line of the Qn versus square root of time plot.
Reviewer 3 Report
The manuscript entitled A validated IVRT method to assess topical creams
containing metronidazole using a novel approach presents a novel method for assessing potential differences between different topical formulations containing metronidazol.
In my opinion, the authors clearly presented the method and the results on the basis of scientific background.
Aminor observation is that the authors use MTZ abreviation before stating what it means and it is not consistent in the text.
Also, I would suggest that the authors introduce the readers about the need of developing a method for assessing differences between topical formulations specifically for metronidazol.
Round 2
Reviewer 1 Report
No comments